# Genetic and Physiological Characterization of Soybean-Nodule-Derived Isolates from Bangladeshi Soils Revealed Diverse Array of Bacteria with Potential Bradyrhizobia for Biofertilizers

**DOI:** 10.3390/microorganisms10112282

**Published:** 2022-11-17

**Authors:** Md Firoz Mortuza, Salem Djedidi, Takehiro Ito, Shin-ichiro Agake, Hitoshi Sekimoto, Tadashi Yokoyama, Shin Okazaki, Naoko Ohkama-Ohtsu

**Affiliations:** 1United Graduate School of Agricultural Science, Tokyo University of Agriculture and Technology (TUAT), Saiwai-cho 3-5-8, Fuchu-shi, Tokyo 183-8509, Japan; 2Institute of Food and Radiation Biology, Atomic Energy Research Establishment, Bangladesh Atomic Energy Commission, Ganakbari, Savar, Dhaka 1207, Bangladesh; 3Faculty of Agriculture, Tokyo University of Agriculture and Technology (TUAT), Saiwai-cho 3-5-8, Fuchu-shi, Tokyo 183-8509, Japan; 4Institute of Global Innovation Research, Tokyo University of Agriculture and Technology (TUAT), Harumi-cho 3-8-1, Fuchu-shi, Tokyo 183-8509, Japan; 5Faculty of Agriculture, Utsunomiya University, Utsunomiya-shi, Tochigi 321-8505, Japan; 6Institute of Agriculture, Tokyo University of Agriculture and Technology (TUAT), Saiwai-cho 3-5-8, Fuchu-shi, Tokyo 183-8509, Japan; 7Faculty of Food and Agricultural Sciences, Fukushima University, Kanayagawa 1, Fukushima-shi, Fukushima 960-1248, Japan

**Keywords:** Bangladesh soil, phylogenetic analysis, *Bradyrhizobium*, soybean, nitrogen-fixation, biofertilizer

## Abstract

Genetic and physiological characterization of bacteria derived from nodules of leguminous plants in the exploration of biofertilizer is of paramount importance from agricultural and environmental perspectives. Phylogenetic analysis of the 16S rRNA gene of 84 isolates derived from Bangladeshi soils revealed an unpredictably diverse array of nodule-forming and endosymbiotic bacteria—mostly belonging to the genus *Bradyrhizobium*. A sequence analysis of the symbiotic genes (*nifH* and *nodD1*) revealed similarities with the 16S rRNA gene tree, with few discrepancies. A phylogenetic analysis of the partial *rrn* operon (16S-ITS-23S) and multi-locus sequence analysis of *atpD*, *glnII*, and *gyrB* identified that the *Bradyrhizobium* isolates belonged to *Bradyrhizobium diazoefficiens*, *Bradyrhizobium elkanii*, *Bradyrhizobium liaoningense* and *Bradyrhizobium yuanmingense* species. In the pot experiment, several isolates showed better activity than *B. diazoefficiens* USDA110, and the Bho-P2-B2-S1-51 isolate of *B. liaoningense* showed significantly higher acetylene reduction activity in both *Glycine max* cv. Enrei and Binasoybean-3 varieties and biomass production increased by 9% in the Binasoybean-3 variety. Tha-P2-B1-S1-68 isolate of *B. diazoefficiens* significantly enhanced shoot length and induced 10% biomass production in Binasoybean-3. These isolates grew at 1–4% NaCl concentration and pH 4.5–10 and survived at 45 °C, making the isolates potential candidates for eco-friendly soybean biofertilizers in salty and tropical regions.

## 1. Introduction

Bacterial diversity analysis is important to understand, maintain, and conserve global genetic resources. Bacteria are not only a vital part of soil and environment, but also the main agents of different nutrient cycling [1]. As new environments and regions are explored, the abundance and fruitfulness of microbial diversity will become increasingly evident [2]. Root nodule bacteria, comprising different types of *Alpha-proteobacteria*, *Beta-proteobacteria*, and possibly *Gamma-proteobacteria*, are of significant importance in biodiversity studies [3]. These soil bacteria are capable of forming root nodules and establishing symbiosis with the roots or stems of leguminous plants [4]. During the symbiotic association process, rhizobia reduce atmospheric nitrogen into ammonium, a usable nitrogen resource for plants, while, concurrently, some compounds are exchanged between the bacteroid and its plant host [5]. The ability of rhizobia to fix nitrogen has a significant effect on reducing the utilization of chemical nitrogen fertilizers in agriculture. The biodiversity of rhizobia represents a valuable bio-resource for the exploration of bacterial strains with suitable traits that can increase legume production [6].

At present, all symbiotic nitrogen-fixing bacteria are categorized in the vast phylum *Proteobacteria*, within the classes *Alphaproteobacteria* (α-rhizobia), *Beta-proteobacteria* (β-rhizobia), and possibly *Gamma-proteobacteria* (γ-rhizobia), with approximately 180 nodulating species in 21 genera [3,7,8]. The symbiotic bacteria in the class *Alpha-proteobacteria* are the most common nitrogen-fixing bacteria, which are distributed in 16 genera of seven families: *Agrobacterium*, *Allorhizobium*, *Ensifer* (previously *Sinorhizobium*), *Neorhizobium*, *Pararhizobium*, *Rhizobium*, and *Shinella* of the family *Rhizobiaceae*; *Aminobacter*, *Phyllobacterium*, and *Mesorhizobium* of *Phyllobacteriaceae*; *Bradyrhizobium* of *Bradyrhizobiaceae*; *Microvirga* and *Methylobacterium* of *Methylobacteriaceae*; *Ochrobactrum* of *Brucellaceae*; *Devosia* of *Hyphomicrobiaceae*; and *Azorhizobium* of *Xanthobacteraceae*. All are members of the order *Rhizobiales*, in which some of the other families are *Bartonellaceae*, *Beijerinckiaceae*, *Cohaesibacteraceae*, *Methylocystaceae*, *Rhodobiaceae*, and *Roseiarcaceae* [3]. Compared with the rhizobia in the class *Alpha-proteobacteria*, the symbiotic bacteria in *Beta-proteobacteria* and *Gamma-proteobacteria* were recognized much later [9,10] and are less diverse, including approximately 20 species in four genera: *Cupriavidus, Paraburkholderia*, and *Trinickia*, belonging to the family *Burkholderiaceae* [11], and *Herbaspirillum*, in the family *Oxalobacteraceae* [9].

All these bacteria form symbioses with different types of leguminous plants. One of those is soybean (*Glycine max* L. Merr.), which belongs to the tribe of *Phaseoleae*, sub-tribe of *Glycininae*, and genus *Glycine*, which originated and was domesticated in China around 5000 years ago [12,13]. The highest diversity of soybean-nodulating rhizobia is found in China and countries in the Americas [12,13,14]. Among rhizobia, *Bradyrhizobium* is the most established soybean microsymbiont and is used as a biofertilizer. The genus *Bradyrhizobium* was established in early 1982, and, to date, 73 species have been recognized worldwide [15,16]. Of the 73 species, *B. elkanii*, *B. japonicum*, *B. diazoefficiens*, *B. daqingense*, *B. liaoningense*, *B. huanghuaihaiense*, and *B. ottawaense* are seven slow-growing rhizobial species that have been described to nodulate soybean [3]. Some species of the genus *Sinorhizobium* (syn. *Ensifer*) nodulate soybean [3], whereas *Mesorhizobium tianshanense*, formerly known as *Rhizobium tianshanense*, from the genus *Mesorhizobium*, has also been reported to be a soybean microsymbiont [17].

Root nodule bacterial diversity associated with soybean (*Glycine max*) is immensely important because soybean is one of the most important legume crops in the world, representing 50% of global area planted with crop legumes and 68% of global legume crop production [7,12,18]. The characterization of soybean-associated bacteria from unexplored regions and environments can aid in the discovery of novel bacteria with potential biofertilizer activity. The application of local or indigenous bacteria associated with soybeans can increase crop productivity more than foreign inoculants because of their adaptation to the local environment and compatibility with local soybean varieties. Thus, the isolation of suitable and potential local bacteria, by analyzing the diversity of soybean root nodule bacteria from Bangladeshi soil samples, can facilitate the development of efficient biofertilizers for soybean.

Moreover, the study of bacterial diversity in Bangladeshi soils associated with soybean is scarce. Therefore, the objectives of this study are as follows:

(i) To achieve a fundamental understanding of the diversity and taxonomic identity of rhizobia associated with soybean root nodules from Bangladeshi soil.

(ii) To estimate the stress tolerance of the isolates and identify suitable stress-tolerant bacteria.

## 2. Materials and Methods

### 2.1. Collection of Soil Samples

Soil samples were collected from 11 districts of different agro-ecological zones in Bangladesh, focusing mainly on areas where soybeans have been cultivated (Table 1 and Figure 1). For each location, soil sample was collected from a depth of ~10–15 cm from different positions of a field and mixed. The soil samples were stored at 4 °C until use.

### 2.2. Collection of Nodules Using Soil Samples in Pot Experiment

The seeds of soybean cultivar *Glycine max* c.v. Enrei (Japanese variety) were surface-sterilized by immersion in 70% ethanol for 30 s, and then in 3% sodium hypochlorite solution for 3 min, and the seeds were washed exhaustively with sterile water [19]. For seed germination, the sterilized seeds were incubated for two days at 28 °C. From each soil sample, 1 g of soil was suspended in 5 mL of sterilized water and soil suspensions were used as inoculants. Each inoculant was applied to 300 mL glass jars containing germinated seeds and sterilized vermiculite, and the jars were placed in growth chamber and kept under controlled conditions with 12 h light/dark photoperiod at temperatures of 25 °C and 18 °C in the light and dark hours, respectively [19]. Plant growth was supported by adding a sterilized N-free nutrient solution [20] to the jars. After four weeks, whole plants were uprooted from the jars, washed in running tap water to remove vermiculite, and the nodules were harvested. Root nodules were surface-sterilized by immersion in 70% ethanol for 30 s and in 3% sodium hypochlorite for 3 min, then washed five times with sterile water [19]. Each nodule was crushed in 100 μL of glycerol solution (15%, *v*/*v*) to obtain a turbid suspension. An aliquot (10 μL) of suspension was plated onto 1.5% yeast extract mannitol agar (YEM) [21] plates and incubated for 3–7 days at 28 °C. The remaining suspension was maintained at −30 °C. Single colonies were selected and checked for purity by repeated streaking onto fresh YEM medium. The isolates were maintained for the long-term into 50% glycerol stocks at −80 °C and the short-term into slant stocks at 4 °C.

### 2.3. Temperature Tolerance Test of Isolates

The isolates used in this study were examined for growth under high-temperature conditions. For the temperature tolerance test, isolates were inoculated on YMA plates and incubated at 10 °C, 20 °C, 28 °C, 37 °C, 40 °C, and 45 °C for 3–7 days [22]. After incubation, bacterial growth was examined.

### 2.4. pH Tolerance Test of Isolates

The isolates used in this study were examined for growth at different pHs. For pH tolerance tests, isolates were inoculated on YMA plates at pH 4.5, 6, 7, 8.5, and 10. The pH was adjusted to pH 4.5 and 6 using 0.5 M HCl, while the pH was adjusted to pH 7, 8.5, and 10 using 0.5 M NaOH. After bacterial inoculation, the plates were incubated at 28 °C for 3–7 days [22]. After incubation, bacterial growth was examined.

### 2.5. Salinity Tolerance Test of Isolates

The isolates used in this study were examined for growth under high-salinity conditions. For the salt tolerance test, isolates were inoculated on YMA plates containing 1%, 2%, 3%, and 4% (*w*/*v*) NaCl concentration and incubated at 28 °C for 3–7 days [22]. After incubation, bacterial growth was examined.

### 2.6. DNA Extraction, Amplification and Sequencing

Genomic DNA for each isolate was extracted from YEM broth culture following the standard protocol using the Wizard^®^ Genomic DNA Purification Kit (WI 53711-5399; Promega Corporation, Madison, WI, USA). DNA sequences corresponding to the 16S rRNA, *rrn* operon (16S-ITS-23S), *nodD1*, *nifH*, *atpD, glnII*, and *gyrB* genes were amplified by PCR using the KOD-Plus-Neo enzyme system (Toyobo Co., Ltd., Osaka, Japan) and sequenced using an Applied Biosystems 3500 Genetic Analyzer using the BigDye^®^ Terminator v3.1 Cycle Sequencing Kit Protocol (Thermo Fisher Scientific, Life Technologies Corporation, California, CA, USA) with the primer pairs shown in Appendix A [23,24,25,26,27,28].

### 2.7. Analysis of DNA Sequences and Phylogeny Genomic

An analysis and quality check of DNA sequences were performed primarily using Sequence Scanner 2.0 software (Thermo Fisher Scientific) and 4Peaks (Nucleobytes). Sequences derived from forward and reverse primers were individually joined by identifying the overlapping sequence between them, using GENETYX version 11 software (Genetyx Corp., Tokyo, Japan) and the online tool Merger (http://www.bioinformatics.nl/cgi-bin/emboss/merger; accessed on 17 June 2019). Multiple sequence alignments of nucleotide sequences and bootstrapping to create maximum likelihood and neighbor-joining phylogenetic trees were performed using MEGA X [29]. The evolutionary history of all phylogenetic trees in this study was inferred by the maximum likelihood method using MEGA X software [29]. The percentage of trees in which the associated taxa clustered together is shown next to the branches. The initial tree(s) for the heuristic search were obtained by applying the neighbor-joining method to a matrix of pairwise distances, estimated using the maximum composite likelihood approach. The tree was drawn to scale, with branch lengths measured in terms of the number of substitutions per site [29].

### 2.8. Analysis of Inoculation Effects of Selected Isolates on Plants

Isolates were grown in yeast-extract mannitol (YM) broth for three days at 28 °C, after which the cells were collected by centrifugation. The cells were then washed and diluted with sterile, ultrapure Milli-Q water. The absorbance of the diluted cells was measured using a spectrophotometer (Ultrospec 3300 Pro; Amersham Biosciences, Cambridge, UK) at 600 nm. For each isolate, the amount of bacterial inoculation was kept almost equal by maintaining (absorbance at 600 nm × volume of inoculation = 1) the same scale and measuring the colony-forming unit (CFU) by plate count. The concentration of cells in the solution used for inoculation was maintained at approximately 10^8^ cells mL^−1^ [24,25]. Surface-sterilized seeds of *Glycine max* cv. Enrei and Binasoybean-3 (Bangladeshi soybean variety, Appendix A) were sown in 300 mL of sterile vermiculite (Yoshino Gypsum Co., Ltd., Tokyo, Japan) in glass jars. After sowing, 1 mL of the cell suspension was applied to the seeds. Three replicates were performed for each isolate. Subsequently, the jars were transferred to a growth chamber. Sterilized N-free nutrient solution [20] was added to the glass jars. The moisture content was maintained at ~70% throughout the growth period. Plants were grown in a growth room under a 12 h light (25 °C)/dark (18 °C) photoperiod. After five weeks of incubation, whole plants were removed and washed from the vermiculite and different parameters, e.g., shoot length, root length, nodule color, nodule size and number, shoot weight, root weight, nodule weight, and ARA, were measured [23,30]. Experiments were performed using a randomized complete block design with 2 soybean varieties, 14 bacteria and 3 replicates for each treatment [31].

### 2.9. Acetylene Reduction Assay

For the acetylene reduction assay (ARA), whole plant roots with root nodules from each tested plant were introduced into a 300 mL airtight jar. Subsequently, 30 mL of air from the jar was replaced with 30 mL of acetylene, and the whole roots with root nodules were further incubated for 1 h at 30 °C to evaluate nitrogenase activity in the root nodules [19]. The amount of ethylene produced owing to the activity of nitrogenase on acetylene enclosed in the jar was determined using a Shimadzu GC-8A gas chromatograph (Shimadzu, Tokyo, Japan) equipped with a Porapak N column (Chrompack, Middelburg, the Netherlands). The roots of plants without bacterial inoculation were used as controls [19].

### 2.10. Nucleotide Sequence Accession Numbers

DNA sequences were deposited in the DNA Data Bank of Japan (DDBJ) under the accession numbers LC652665 to LC652748 for the 16S rRNA gene, LC658965 to LC658985 for 16S rRNA, 16S-23S rRNA internal transcribed spacer, tRNA-Ile, tRNA-Ala, 23S rRNA gene, LC670617 to LC670629 for *atpD*, LC670630 to LC670642 for *glnII*, LC670643 to LC670655 for *gyrB*, LC670656 to LC670720 for *nifH*, and LC671364-LC671428 for *nodD1*.

### 2.11. Statistical Analysis

The experimental data obtained from plant tests were subjected to statistical analyses, such as multiple comparison using general linear model with Dunnett’s Post Hoc test (with 95% confidence level), a correlation analysis using the bivariate correlations method with Pearson Correlation Coefficient and two-tailed test of significance, and regression analysis using the linear regression method with 95% confidence intervals level using IBM SPSS Statistics, Version 23.0.

## 3. Results

### 3.1. Isolation of Bacteria from Root Nodule

Soil samples from 11 districts of Bangladesh were used for root-nodule bacterial isolation. Soybean (*Glycine max* cv. Enrei) plants were grown in vermiculite using the soil samples. A total of 84 isolates were obtained from the selected root nodules of soybean plants.

### 3.2. Phylogenetic Analysis Based on the 16S rRNA Genes

To determine the phylogenetic position of the isolates, 16S rRNA gene sequencing and phylogenetic analysis with highly similar reference strains were performed using a 1348 nt sequence for each bacterium.

Subsequently, the 84 isolates were classified into eight groups (Figure 2a) belonging to the *Alpha-proteobacteria*, *Beta-proteobacteria*, *Gamma-proteobacteria*, *Firmicutes*, and *Actinobacteria* phylum/class. The majority of isolates (65/84; 77.4%), showed a close relationship with the genus *Bradyrhizobium*, belonging to the class *Alphaproteobacteria* and family *Rhizobiales*. The *Bradyrhizobium* isolates were classified into three groups: A (16S), B (16S), and C (16S). Forty-five isolates showing a close relationship with *Bradyrhizobium diazoefficiens* were grouped in A (16S) (Figure 2b). Group B (16S), consisting of 16 isolates, was found to belong to the same clade as *Bradyrhizobium liaoningense* and *Bradyrhizobium yuanmingense* (Figure 2c). Four isolates belonging to group C (16S) were observed to have the highest similarity to *Bradyrhizobium elkanii*. Group D (16S), with six isolates, showed a close relationship with the genus *Methylobacterium*, which also belongs to the class *Alpha-proteobacteria* and family *Rhizobiales.*

Two isolates of group E (16S) and seven isolates of group F (16S) were observed in the same branch as the genus *Stenotrophomonas* of the *Gamma-proteobacteria* class and genus *Pandoraea* of the *Beta-proteobacteria* class, respectively. Group G (16S), comprising only one isolate, was observed to reside in the same branch as *Leifsonia lichenia* of the *Actinobacteria* class. Three isolates of group H (16S) belonged to the genus *Bacillus* and *Firmicutes*.

### 3.3. Phylogenetic Analysis Based on the nifH Gene

A total of 65 isolates and seven closely similar type strains were phylogenetically characterized based on 718 nt long DNA fragments from the nifH region. As shown in Figure 3a, the isolates were classified into three groups. Group A (*nifH*) comprised the majority of the isolates (59/65) positioned in the same branch as the reference strains of *B. diazoefficiens* (Figure 3b). Isolates of group B (*nifH*) were spotted in the same clade as the strains of *B. yuanmingense*. Four isolates from group C (*nifH*) resembled the *B. elkanii* strains.

### 3.4. Phylogenetic Analysis Based on the nodD1 Gene

A total of 65 isolates and eight reference type strains were phylogenetically classified based on 353 nt long DNA fragments from the *nodD1* region. As shown in Figure 4, four groups were identified. Group A (*nodD*) consisted of 37 isolates clustered in the same branch as the *B. diazoefficiens* strains, with *B. liaoningense* in the neighboring branch, and 21 isolates of group B (*nodD*) were spotted in the same branch as *B. ottawaense*. Group C (*nodD*) and group D (*nodD*), containing one and four isolates, were formed with type strains of *B. yuanmingense* and *B. elkanii*, respectively, which was almost similar to the results of previous phylogenetic trees (Figure 2 and Figure 3). One of the isolates, Tan-P1-B2-St1-28, which showed similarity with *B. yuanmingense* in the 16S rRNA- and *nifH*-based trees, and Din-P2-M1-M1-25 isolate, a sequence similarity of ~83% (100% query coverage) with *B. diazoefficiens*, were found in a separate branch, with *B. diazoefficiens*, *B. ottawaense*, and *B. liaoningense* strains in the neighboring branches.

### 3.5. Selection of Isolates for Further Genetic and Physiological Characterization

From the 16S rRNA sequence analysis, it was found that, among the 84 isolates, 71 were from *Alpha-proteobacteria*, seven from *Beta-proteobacteria*, two from *Gamma-proteobacteria*, three from *Firmicutes*, and one from *Actinobacteria* (Appendix A). Of the 84 isolates from 11 districts/locations in Bangladesh, 13 representative *Bradyrhizobium* isolates were selected based on location and the phylogenetic analysis of the 16S rRNA, *nifH*, and *nodD1* genes for pot experiments, as well as further genetic and physiological characterization.

### 3.6. Phylogenetic Analysis of the rrn Operon (16S-ITS-23S)

Phylogenetic analysis of the partial *rrn* operon (16S-ITS-23S) was performed by concatenating the almost complete 16S rRNA gene, full-length 16S-23S internal transcribed spacer (ITS) region, and partial 23S rRNA gene sequences of the selected isolates (Figure 5). The length of the sequences of the isolates was 2610-2685 nt. The selected isolates and seven reference type strains were phylogenetically characterized using the maximum likelihood method, and four groups were obtained. Group A (16S-ITS-23S) comprised seven isolates that branched with *B. diazoefficiens.* Three isolates from group B (16S-ITS-23S) were closely related to *B. liaoningense*, and one isolate from group C (16S-ITS-23S) was closely related to *Bradyrhizobium arachidis* and *Bradyrhizobium guangxiense*. Two isolates clustered with *B. elkanii* in Group D (16S-ITS-23S).

### 3.7. Multi-Locus Sequence Analysis of the Housekeeping Genes

Multi-locus sequence analysis (MLSA) was performed by concatenating the almost complete *atpD*, *glnII*, and *gyrB* genes (total of 2559 nt length of sequence of each isolate) of selected *Bradyrhizobium* isolates with reference strains using the maximum likelihood method (Figure 6). The selected isolates and nine reference strains were phylogenetically characterized using the maximum likelihood method, and four groups were obtained.

Group A (MLSA) comprised of seven isolates that branched with *B. diazoefficiens*. Three isolates from Group B (MLSA) and one isolate from Group C (MLSA) were closely related to *B. liaoningense* and *B. yuanmingense*, respectively. Two isolates clustered with *B. elkanii* in Group D (MLSA).

### 3.8. Symbiotic Performances

To assess the symbiotic ability of the 13 selected isolates, pot experiments were performed under controlled laboratory conditions using two soybean varieties, *Glycine max* cv. Enrei and Binasoybean-3. Different parameters, such as shoot length, main root length, nodule color (Appendix A), nodule size and number, shoot weight, root weight, nodule weight, and ARA, were measured.

The data of nodule number of different sizes, (e.g., medium size (2–5 mm) and small size (<2 mm)) of both varieties inoculated with bacteria are presented in Table 2. Isolates Bho-P2-B2-S1-51 and Mym-P2-M3-S1-45 produced the highest number of medium- and small-sized nodules in Enrei plants, respectively. Similar to the Enrei variety, in the case of Binasoybean-3, Bho-P2-B2-S1-51 produced the highest number of medium-sized nodules, whereas Din-P2-M1-M1-25 produced the highest number of small-sized nodules. The total number of nodules produced was higher in the Binasoybean-3 variety than that in the Enrei variety.

The shoot and root lengths of soybean plants (Enrei variety) inoculated with selected 13 isolates and *B. diazoefficiens* USDA110 are shown in Appendix A. Most of the isolated bacteria (11/13) enhanced shoot growth in inoculated plants in comparison to *B. diazoefficiens* USDA110, and Lax-P1-M1-S1-46 isolates increased shoot growth and Bog-P3-B1-S1-29 demonstrated the highest root length development, but none of the isolates produced statistically significantly higher shoot or root growth than *B. diazoefficiens* USDA110.

In the Binasoybean-3 variety, most of the isolates (10/13) enhanced shoot growth in inoculated plants compared to *B. diazoefficiens* USDA110. However, among them, only the Tha-P2-B1-S1-68 isolate showed significantly higher shoot length than *B. diazoefficiens* USDA110 (Appendix A). In the case of root length, no significant increase was observed in comparison with *B. diazoefficiens* USDA110, and Bog-P3-B1-S1-29 produced the highest root length among all bacteria.

The data on shoot dry weight (DW), root DW, and nodule DW of soybean plants inoculated with isolates and *B. diazoefficiens* USDA110 are presented in the bar chart in Appendix A. With regard to the soybean Enrei variety, most of the plants inoculated with isolated bacteria accumulated similar or higher amounts of shoot dry weight in comparison to *B. diazoefficiens* USDA110. However, in the case of root DW and nodule DW, only a few isolates produced higher amounts (Appendix A). None of the isolates showed significantly higher shoot, root, or nodule DW gain than *B. diazoefficiens* USDA110. Din-P2-M1-M1-25, Bog-P3-B1-S1-29, and Nat-P3-M1-S1-79 isolates stimulated the highest shoot DW, root DW, and nodule DW, respectively.

The shoot DW, root DW, and nodule DW of the Binasoybean-3 variety measured from pot experiments are displayed in Appendix A. Compared to *B. diazoefficiens* USDA110, a slight increase in shoot DW yield was observed in plants inoculated with some isolates (6/13). Pan-P1-B1-S1-69 isolate stimulated the highest shoot DW. In contrast to the Enrei variety, more isolates induced a higher root and nodule DW in the Binasoybean-3 variety and the Nil-P2-B1-M1-36 isolate produced a significantly higher amount of nodule DW in comparison to *B. diazoefficiens* USDA110 (Appendix A). This isolate also produced the highest root DW.

The results of biomass production (shoot + root + nodule DW) and ARA activity in the Enrei variety and Binasoybean-3 inoculated with isolated bacteria and *B. diazoefficiens* USDA110 are shown in Table 3. In the case of the Enrei variety, no substantial difference in biomass production was observed between plants inoculated with isolates and *B. diazoefficiens* USDA110. Some isolates produced higher amounts of plant biomass in comparison with *B. diazoefficiens* USDA110, and Din-P2-M1-M1-25 isolate produced the highest plant biomass among isolates, which was 7.4% higher than *B. diazoefficiens* USDA110. Regarding ARA activity, 8 out of 13 isolates demonstrated higher ARA activity than *B. diazoefficiens* USDA110, and Bho-P2-B2-S1-51 showed significantly higher ARA activity than *B. diazoefficiens* USDA110.

For the Binasoybean-3 variety, considerable differences were not observed between plants inoculated with isolates and *B. diazoefficiens* USDA110. Six isolates produced equal or higher amounts of plant biomass in comparison with *B. diazoefficiens* USDA110, and Nil-P2-B1-M1-36 isolate produced the highest plant biomass among all the isolates, which was 14% higher than *B. diazoefficiens* USDA110. Regarding ARA activity, six isolates out of 13 demonstrated higher ARA activity than *B. diazoefficiens* USDA110, and Bho-P2-B2-S1-51 showed significantly higher ARA activity than *B. diazoefficiens* USDA110. Bho-P2-B2-S1-51 was the only isolate that produced significant ARA in both the Enrei and Binasoybean-3 varieties.

### 3.9. Stress Tolerance Tests

Data from the stress tolerance tests (salinity, temperature, and pH) are shown in Table 4. In the salinity tolerance test, all the investigated isolates grew at 1–4% NaCl concentration. In the temperature tolerance test, all the isolates grew at 20 °C, 28 °C, and 37 °C. Interestingly, all the isolates grew at 40 °C, furthermore, the Bho-P2-B2-S1-51, Bog-P3-B1-S1-29, Mym-P2-M3-S1-45, Nat-P3-M1-S1-79, and Tha-P2-B1-S1-68 isolates survived at 45 °C. Some of the isolates grew at 10 °C. In the pH tolerance test, all isolates developed colonies at pH 6.0–10.0. Except for Mym-P3-M2-S1-40 and Noa-P1-B1-M1-31, all isolates also grew at pH 4.5.

## 4. Discussion

The principal objective of this study was to achieve a fundamental understanding of the diversity, physiology, and symbiotic characteristics of nitrogen-fixing bacteria associated with soybean root nodules in Bangladeshi soil samples. To accomplish these goals, experiments related to bacterial isolation from root nodules, the phylogenetic characterization of bacterial strains with the 16S rRNA, *rrn* operon (16S-ITS-23S), *nifH*, *nodD1*, *atpD*, *glnII*, and *gyrB* genes, stress-tolerance tests, and plant tests in pots were performed.

Soil samples were collected from 11 different districts in Bangladesh, focusing on areas where soybean had been cultivated. Soybean plants were grown in pots using suspensions of the soil samples, and 84 isolates were isolated from the nodules of the soybean plants. Phylogenetic analysis of 84 soybean root-nodule isolates using the 16S rRNA gene revealed that the majority of the isolates (65/84) showed a phylogenetic resemblance with the genus *Bradyrhizobium* (Figure 2). Among these isolates, 45 showed a close relationship with *B. diazoefficiens* and *B. japonicum,* 16 with *B. liaoningense* and *B. yuanmingense,* and four with *B. elkanii*. The prevalence of *Bradyrhizobium* strains, especially *B. diazoefficiens* and *B. japonicum*, in root nodules of soybean is a common scenario and is scientifically expected. *B. diazoefficiens* (and/or *B. japonicum*) and *B. elkanii* species have been found in diverse climatic regions across the world, and strains of *B. liaoningense* are abundantly found in alkaline soils of temperate to subtropical climates in South and Southeast Asia, while the warm tropical climates of India and Nepal support *B. yuanmingense* [32]. Furthermore, in a research expedition for the genetic analysis of 76 bacteria isolated from root nodules of soybean inoculated with soil samples from Myanmar, India, Nepal, and Vietnam, Vinuesa et al. (2008) showed that 75 of the isolates were phylogenetically similar to *B. japonicum* (USDA110) or currently, *B. diazoefficiens* (in Nepal), *B. liaoningense* (in Myanmar, Vietnam)*, B. yuanmingense* (in India, Myanmar, and Vietnam)*,* and *B. elkanii* (in Myanmar) [28]. Another study on root-nodule isolates of soybean from five regions in India, performed by Appunu et al. (2008), showed that 36% of isolates were identified as *B. yuanmingense*, 26% as *B. liaoningense*, and 38% were different from all described *Bradyrhizobium* species but showed the same symbiotic genotype as *B. liaoningense* and *B. japonicum* [33].

Besides these slow-growing *Bradyrhizobium* isolates of the *Alpha-proteobacteria* class, some other bacterial strains, such as *Methylobacterium* of *Alpha-proteobacteria*, *Pandoraea* of *Beta-proteobacteria*, *Stenotrophomonas* of *Gamma-proteobacteria*, *Bacillus* of *Firmicutes*, and *Leifsonia* of *Actinobacteria* phylum/class, were also found in this study. *Methylobacterium* of the *Methylobacteriaceae* family and *Rhizobiales* order, belonging to the class *Alpha-proteobacteria*, is a well-established group of bacteria associated with the root nodules of leguminous plants [34,35,36,37]. Bacterial strains of the genus *Burkholderia* of the *Burkholderiaceae* family of *Beta-proteobacteria* have also been isolated by many researchers from root nodules [6,8,38,39]. Therefore, the isolation of bacteria belonging to the *Pandoraea* genus of the *Burkholderiaceae* family of the *Beta-proteobacteria* class is new but not unnatural. *Stenotrophomonas* bacteria of the *Gamma-proteobacteria* class have been isolated from the rhizosphere soil of leguminous plants [40]. Therefore, the existence of bacteria of this genus in root-nodules is also reasonable. Different species of *Bacillus* are commonly found in the root nodules of different leguminous plants and are thus treated as a well-known phenomenon [24,25,41]. In addition, many bacterial strains belonging to the phylum *Actinobacteria*, such as *Frankia*, *Rhodococcus*, *Arthrobacter*, *Brevibacterium*, *Micromonospora*, and *Streptomyces*, have also been isolated from root nodules and characterized for their nitrogen-fixing or plant-growth-promoting capabilities [3,42,43,44,45,46].

According to researchers, the analysis of symbiotic genes, such as *nifH*, *nifD*, *nifK*, *nodA*, *nodB*, *nodC*, and *nodD*, of root nodule bacteria can provide valuable insights into their host range and symbiotic relationships, which may vary from those expected by rRNA-based phylogenies [6,20,38,47]. In the *nifH* phylogenetic analysis, 59 out of 65 *Bradyrhizobium* isolates possessed the *nifH* gene, similar to that of *B. diazoefficiens* (Figure 3). Except for the four isolates that showed 16S rRNA gene resemblance with *B. elkanii* (group C (16S) of Figure 2), and two isolates out of 16 of *Bradyrhizobium liaoningense* and *Bradyrhizobium yuanmingense* (group B (16S) of Figure 2c), all other isolates showed *nifH* gene similarity to *B. diazoefficiens.* In an analysis of a database consisting of 32,954 aligned sequences of the *nifH* gene, Gaby et al. (2014) revealed that genomes that had >97% similarity in the 16S rRNA gene could possess up to 23% dissimilarity in the *nifH* sequences [48]. Therefore, *B. yuanmingense* and *B. liaoningense*, which have a 16S rRNA gene similarity of approximately 99% with *B. diazoefficiens*, can logically possess the *nifH* gene resembling *B. diazoefficiens*, which has approximately 10% sequence dissimilarity. This transfer of the *nifH* gene may be the outcome of vertical gene transfer in the symbiotic region of *Bradyrhizobium* species. In the case of *nodD1* gene phylogeny, almost similar results were obtained for the nifH gene tree, with some discrepancies. Most isolates (*n* = 37) clustered in the same clade as *B. diazoefficiens* (in group A (*nodD*) of Figure 4b). The Tan-P1-B2-St1-28 and Din-P2-M1-M1-25 isolates showed a sequence similarity of ~83% (100% query coverage) with *B. diazoefficiens* and were placed in a different branch adjacent to *Bradyrhizobium* branches. No change was observed in the four isolates that showed similarity with *B. elkanii* in the *nodD1* gene tree or the 16S rRNA and *nifH* tree, suggesting a strong phylogenetic relation with *B. elkanii*. Therefore, further analysis should be performed to verify the symbiotic gene transfer process.

Some researchers have shown that the analysis of the *rrn* operon (16S–ITS–23S) signifies more than the analysis of the 16S rRNA gene for the resolution of taxa at the species level [49,50]. Phylogenetic analysis of the partial *rrn* operon consisting of the almost complete 16S rRNA gene, full-length 16S-23S internal transcribed spacer (ITS) region, and partial 23S rRNA gene sequences for the selected isolates was congruent with the 16S rRNA gene-based tree (Figure 5). MLSA was also performed by concatenating the *atpD*, *glnII*, and *gyrB* genes of selected *Bradyrhizobium* isolates with type strains using the maximum likelihood method (Figure 6). Seven of the 13 isolates branched with *B. diazoefficiens*, three branched with *B. liaoningense*, and one branched with *B. yuanmingense*. Two isolates clustered with *B. elkanii*. For all these isolates, the MLSA position could be considered the true taxonomic identity. The use of MLSA of the housekeeping genes in bacterial species definition and identification has been strongly endorsed by the scientific community [51]. Unless whole-genome sequencing is performed, MLSA is considered a distinctive identification process for exploring the genetic diversity within a proposed species [51]. Therefore, based on 16S rRNA gene, MLSA and *rrn* operon phylogenetic analysis, except for the Tan-P1-B2-S1N-84, the other 12 isolates can be classified as *B. diazoefficiens* sp. Din-P2-M1-M1-25, *B. diazoefficiens* sp. Mym-P2-M3-S1-45, *B. diazoefficiens* sp. Mym-P3-M2-S1-40, *B. diazoefficiens* sp. Nat-P3-M1-S1-79, *B. diazoefficiens* sp. Nil-P2-B1-M1-36, *B. diazoefficiens* sp. Noa-P1-B1-M1-31, *B. diazoefficiens* sp. Tha-P2-B1-S1-68, *B. elkanii* sp. Bog-P3-B1-S1-29, *B. elkanii* sp. Lak-P1-S1-M1-46, *B. liaoningense* sp. Bho-P2-B2-S1-51, *B. liaoningense* sp. Lak-P1-M1-S1-85, *B. liaoningense* sp. Pan-P1-B1-S1-69 for taxonomic identity. For Tan-P1-B2-S1N-84 isolate further analysis is needed to confirm the taxonomic identity.

In this study, among the different phylogenetic methods, including Bayesian inference, maximum-likelihood, maximum-parsimony, and neighbour-joining, the maximum-likelihood method was applied for phylogenetic reconstruction. Although each of these techniques has its strengths and weaknesses, maximum-likelihood (ML) and Bayesian methods typically provide more realistic and robust phylogenies because they employ explicit models of molecular evolution [51]. The analysis of the highest DNA sequence similarity of the selected isolates with *Bradyrhizobium* type strains showed no variation in the case of *Bradyrhizobium diazoefficiens* isolates for all analyzed genes. However, some discrepancies were observed for isolates of other genera (Table 5). Among the housekeeping genes, *gyrB* was found to be more decisive in identifying the isolates than *atpD* and *glnII*, and showed similarities with *rrn* operon sequence data.

To evaluate the symbiotic ability of the 13 selected isolates, pot experiments were performed using two soybean varieties, *Glycine max* cv. Enrei and Binasoybean-3, and different parameters were measured. All isolates formed red or pink (cross-sections) nodules, indicating the presence of leghemoglobin in the soybean Enrei variety and Binasoybean-3 variety, except Tan-P1-B2-S1N-84 belonging to *B. yuanmingense,* which did not form any nodules in the Enrei variety (Appendix A, Table 2). The variation in nodule production could be due to the variation in the agronomic traits of the two soybean cultivars and differences in the compatibility of the isolates with the cultivars. Wang et al. (2019) described that soybean cultivars can influence nodule formation, and some bacteria can form nodules in some varieties, while others cannot [3]. The number of isolates that formed nodules and the number of nodules formed were higher in the Binasoybean-3 variety than those in the Enrei variety (Table 2). This could be due to the fact that Binasoybean-3 is a local Bangladeshi variety, meaning that isolates of Bangladeshi soil samples have a higher affinity towards it.

The performance of the isolates varied in terms of effective symbiosis and biomass production. The nodulation and nitrogen fixation depends on many aspects, especially the symbiotic genes (such as such as *nifH*, *nifD*, *nifK*, *nodA*, *nodB*, *nodC*, and *nod*, etc.), which can vary from rhizobia to rhizobia; thus, different rhizobial species show different host-specificity and nitrogen fixation performances [3,7]. When inoculated, some of the isolates enhanced biomass production in both the Enrei and Binasoybean-3 varieties in comparison to *B. diazoefficiens* USDA110, whereas some decreased. In the case of Enrei, the Din-P2-M1-M1-25 isolate produced the highest plant biomass among isolates, which was 7.4% higher than *B. diazoefficiens* USDA110. For Binasoybean-3, Nil-P2-B1-M1-36 isolate produced the highest plant biomass among all the isolates, which was 14% higher than *B. diazoefficiens* USDA110. In comparison to *B. diazoefficiens* USDA110, three isolates produced 10% or higher biomass gain in Binasoybean-3 variety, compared to no isolate in the case of the Enrei variety. These results suggest that the isolates were more compatible with Bangladeshi local varieties. Having a phylogenetic similarity with *B. liaoningense*, Bho-P2-B2-S1-51 was the only isolate that produced a significant increase in ARA compared to *B. diazoefficiens* USDA110 in both the Enrei and Binasoybean-3 varieties, increasing plant biomass production by 9% in case of Binasoybean-3 variety. Therefore, this isolate could be a potential biofertilizer candidate. Regarding Binasoybean-3, the Tha-P2-B1-S1-68 isolate belonging to *B. diazoefficiens* significantly increased shoot length compared to *B. diazoefficiens* USDA110 (Appendix A) and enhanced biomass production by 10% (Table 3), making the isolate a suitable candidate as biofertilizer for local varieties. According to Chen et al. (2021), the highly effective symbiosis could only be obtained when the host specificity and habitat specificity cooperated in a rhizobial symbiont; therefore, the rhizobia and legume variety of same locality can demonstrate better symbiotic effects [7].The relationships between different parameters of pot experiments, e.g., nodule number, small and medium size nodule, nodule dry weight, ARA activity, and biomass, were observed to be non-linear and varied from isolate to isolate (Table 6). Among the different parameters, it was observed that, for both soybean varieties, nodule dry weight had a significant positive correlation with ARA (in µmol/h/plant) and biomass DW, suggesting that with an increase in nodule dry weight, ARA (in µmol/h/plant) and biomass production may increase. It is known that nodule dry weight is a good indicator of symbiotic efficacy, and an important parameter in strain evaluation [52]. Allito et al. also observed similar correlation and mentioned nodule dry weight as one of the key indicators of effective legume-rhizobia symbiosis [53]. Although the medium-size nodule number showed a significantly high positive correlation with biomass DW and ARA (in µmol/h/plant) in Binasoybean-3, small nodule numbers displayed insignificant and/or negative correlations, suggesting that a larger nodule size can significantly contribute to biomass production and ARA (in µmol/h/plant) than immature/smaller ones. No significant positive correlations were observed with other parameters. Similar to these results, some researchers did not find any clear relationship between ARA activity, nodule numbers and plant biomass in their studies [25,54].

In the stress tolerance tests, all the selected isolates grew at 1–4% NaCl concentration. Similarly, most isolates grew at pH 4.5, except four isolates, and almost all isolates grew at pH 4.5–10, which demonstrate that the isolates are tolerant to high salt and pH conditions. Under saline conditions, salt-tolerant rhizobia are beneficial for host plant growth. For instance, the growth of alfalfa is enhanced when inoculated with the salt-tolerant *Rhizobium* strain Rm1521 rather than the salt-sensitive strain A2 under NaCl treatment [55]. Among the isolates, five isolates, namely, Bho-P2-B2-S1-51, Bog-P3-B1-S1-29, Mym-P2-M3-S1-45, Nat-P3-M1-S1-79, and Tha-P2-B1-S1-68, survived at 45 °C, demonstrating their suitability as biofertilizer candidates for tropical environments. Chen et al. (2002) isolated Rhizobia from soybeans that could grow at 40 °C [56]. Correspondingly, *Phaseolus vulgaris* nodule bacteria were found to survive at 47 °C [57], and *Rhizobium* isolated from *Sesbania aculeata* plants were observed to grow at 50 °C [58,59]. As temperature, salinity, origin of cultivar, etc., are prominent effectors of rhizobial symbiosis [19], the isolates with temperature and salinity tolerance could be potential biofertilizers, especially for local legume varieties.

## 5. Conclusions

This study represents the first genetic and physiological characterization of soybean-nodulating isolates from different regions of Bangladesh, one of the top soybean oil consumer countries of the world. Surprisingly, diverse arrays of both nodule-forming and endosymbiotic bacteria belonging to *Bradyrhizobium* and *Methylobacterium* of *Alphaproteobacteria*, *Pandoraea* of *Beta-proteobacteria*, *Stenotrophomonas* of *Gamma-proteobacteria*, *Bacillus* of *Firmicutes*, and *Leifsonia* of *Actinobacteria* phylum/class were found to be present in the root nodules of soybean plants grown in pots using a suspension of soil samples from Bangladesh, with some potential isolates showing a higher symbiotic ability than *B. diazoefficiens* USDA110 in pot experiments. One of the selected isolates, Bho-P2-B2-S1-51, which showed a phylogenetic similarity to *B. liaoningense*, produced a significant increase in ARA compared to *B. diazoefficiens* USDA110 in both the Enrei and Binasoybean-3 varieties, induced plant biomass production by 9% in case of Binasoybean-3 variety, and survived at 45 °C. Therefore, this isolate could be a potential biofertilizer candidate. With regard to Binasoybean-3, the Tha-P2-B1-S1-68 isolate, which showed a phylogenetic similarity to *B. diazoefficiens*, also significantly increased shoot length compared to *B. diazoefficiens* USDA110, enhanced biomass production by 10%, and could grow at 45 °C, making this isolate a suitable candidate for use as biofertilizer for local Bangladeshi soybean varieties. Further studies, such as whole-genome sequence analysis and field trials, will need to be performed for prospective candidate isolates as biofertilizers to acquire an in-depth understanding of the genetic make-up and functional characteristics of soybean-rhizobium symbiosis.

## Figures and Tables

**Figure 1 microorganisms-10-02282-f001:**
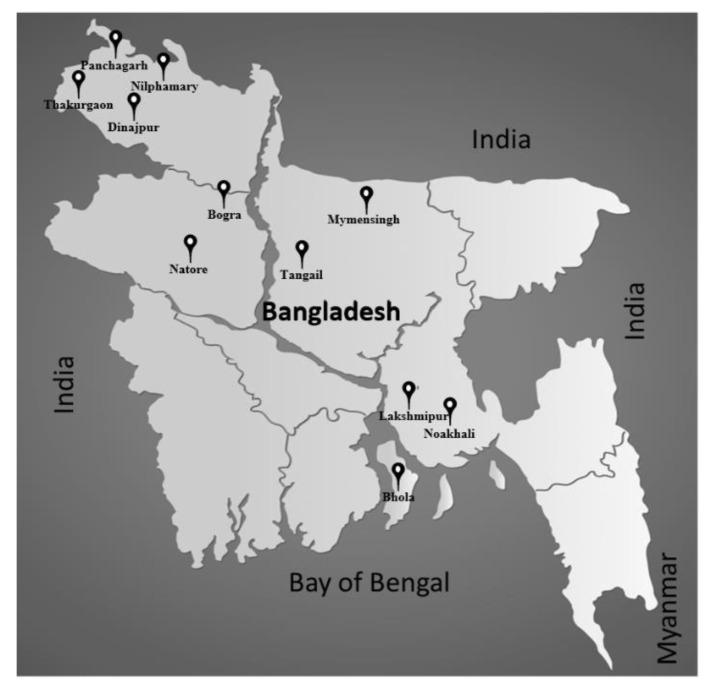
Map of soil sample collection sites of Bangladesh.

**Figure 2 microorganisms-10-02282-f002:**
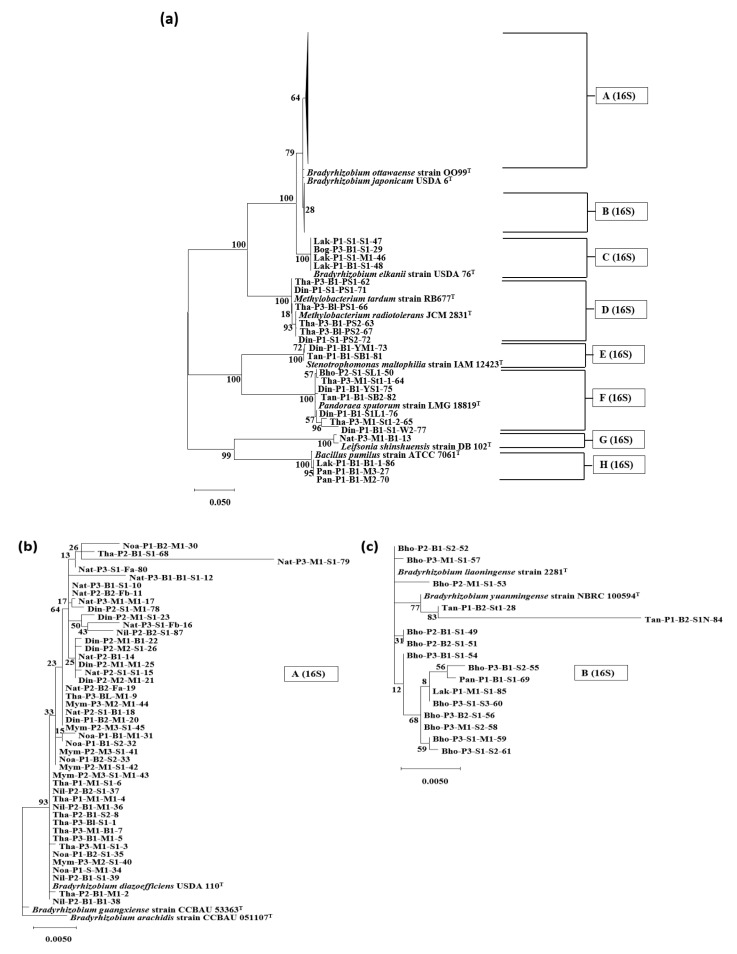
Phylogenetic analysis of the 16S rRNA gene of 84 isolated bacteria with reference strains. The percentage of trees in which the associated taxa clustered together after 1000 Bootstrap replications is shown next to the branch points. (**a**) The main tree with two compressed sub-trees is shown with (**b**,**c**) the respective sub-trees.

**Figure 3 microorganisms-10-02282-f003:**
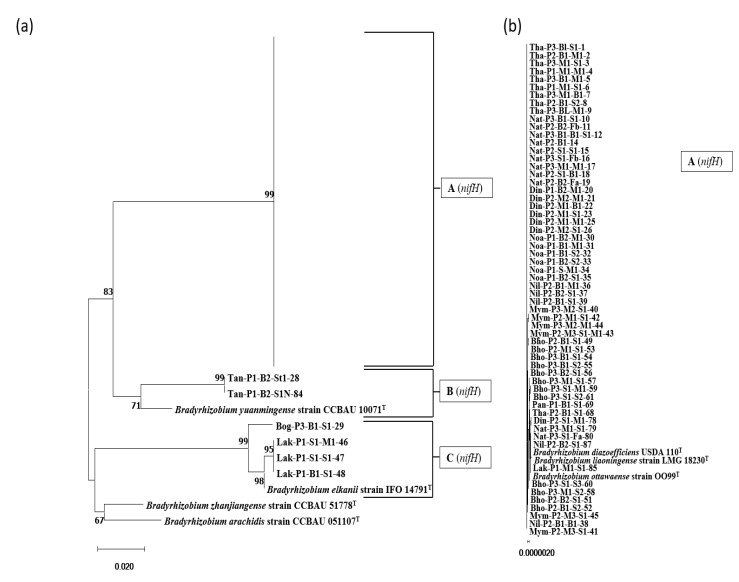
Phylogenetic analysis of the *nifH* gene of isolated bacteria with reference strain. The percentage of trees in which the associated taxa clustered together after 1000 Bootstrap replications is shown next to the branch points. (**a**) Main tree with one compressed sub-tree has been shown with (**b**) the respective sub-tree.

**Figure 4 microorganisms-10-02282-f004:**
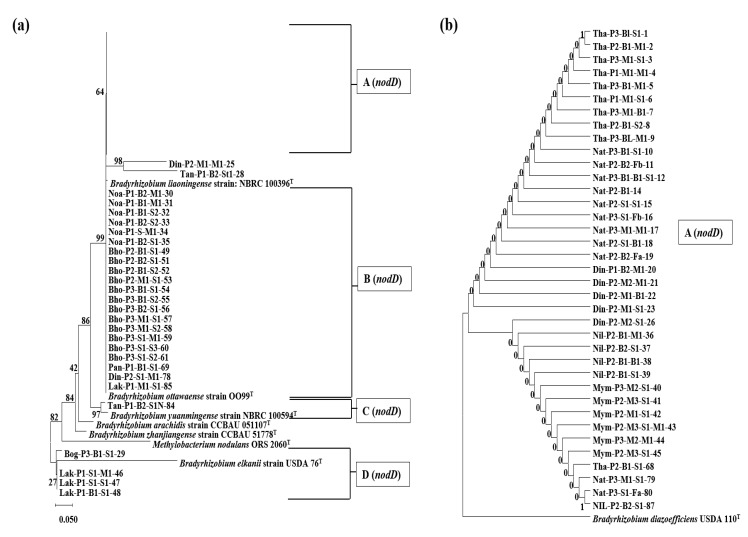
Phylogenetic analysis of the *nodD1* gene of isolated bacteria with reference strain. The percentage of trees in which the associated taxa clustered together after 1000 Bootstrap replications is shown next to the branch points. (**a**) Main tree with one compressed sub-tree has been shown with (**b**) the respective sub-tree.

**Figure 5 microorganisms-10-02282-f005:**
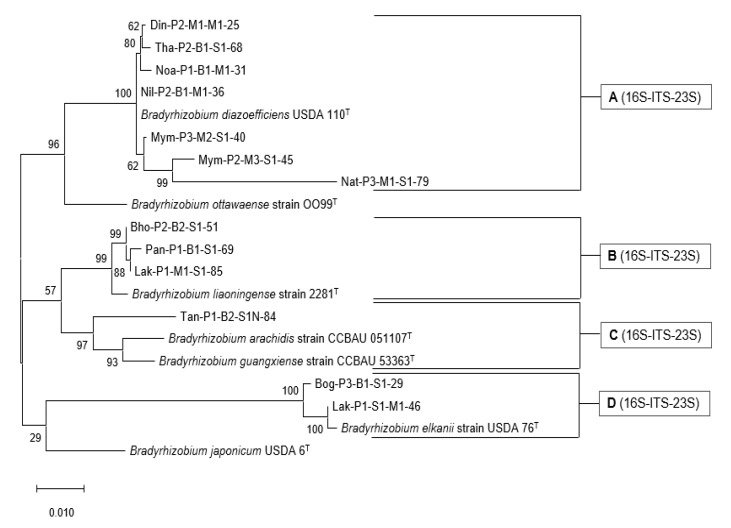
Sequence analysis of the *rrn* operon (16S-ITS-23S) of selected bacteria with reference strains. The percentage of trees in which the associated taxa clustered together after 1000 Bootstrap replications is shown next to the branch points.

**Figure 6 microorganisms-10-02282-f006:**
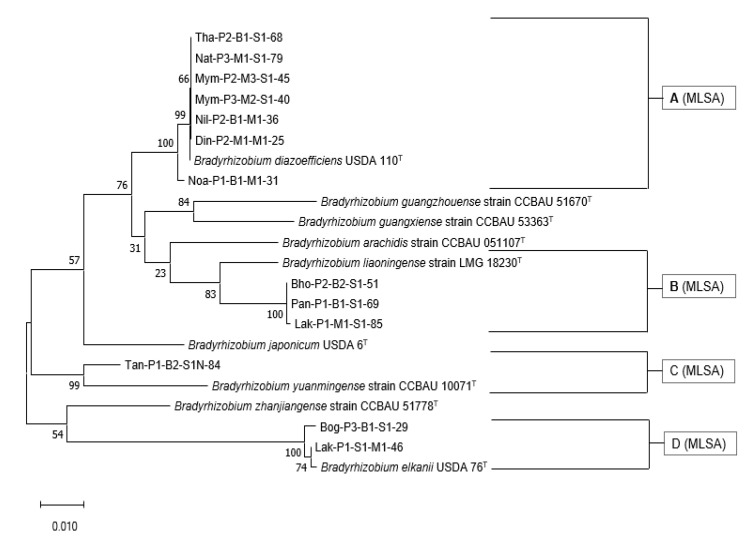
Multi-locus sequence analysis of the *atpD*, *glnII*, and the *gyrB* gene of selected *Bradyrhizobium* isolates with reference strains. The percentage of trees in which the associated taxa clustered together after 1000 Bootstrap replications is shown next to the branch points.

**Table 1 microorganisms-10-02282-t001:** Information of soil sample collection sites.

No.	Name of Site	Collection Date (dd/mm/yy)	Soil Type	Crop History	Soil pH	Latitude	Longitude	Soybean Cultivation
1	Bhola	4 February 2018	Clay loam	Soybean–rice–rice–soybean–rice–rice	6.85	22.6883	90.5975	Yes
2	Bogra	2 February 2018	Loamy	Mustard–rice–rice	6.92	24.8881	89.3869	No
3	Dinajpur	5 February 2018	Sandy loam	Soybean–rice–potato–vegetable	6.30	25.8373	88.4794	Yes
4	Lakshmipur	6 February 2018	Sandy loam	Soybean–rice–soybean–rice–so–bean	6.19	22.7097	90.9952	Yes
5	Mymensingh	4 February 2018	Loamy	Soybean–soybean–soybean	6.49	24.7244	90.4300	Yes
6	Natore	2 February 2018	Loamy	Soybean–rice–soybean–rice	6.41	24.3720	88.8991	Yes
7	Nilphamary	28 January 2018	Sandy loam	Corn–rice–jute–soybean	6.87	26.1102	88.8506	Yes
8	Noakhali	6 February 2018	Sandy loam	Soybean–rice–soybean–rice–soybean	6.68	22.7388	91.0677	Yes
9	Panchagarh	5 February 2018	Sandy loam	Soybean–rice–groundnut–rice–groundnut	6.04	26.2440	88.5598	Yes
10	Tangail	8 February 2018	loamy	Mustard–jute–rice–mustard–jute–rice	6.04	24.2341	89.8598	No
11	Thakurgaon	3 February 2018	Loamy	Mustard–rice–soybean–rice	6.39	25.9208	88.4702	Yes

**Table 2 microorganisms-10-02282-t002:** Data of medium size (2–5 mm) and small size (<2 mm) nodule number of soybean Enrei variety and Binasoybean-3 variety Inoculated with selected isolates and *B. diazoefficiens* USDA110. All experiments were performed in triplicate and the data are presented as the mean ± STDEV.

	Enrei Variety		Binasoybean-3 Variety	
Bacteria Name	Medium Size Nodule	Small Size Nodule	Medium Size Nodule	Small Size Nodule
USDA110	10.00 ± 1.00	9.00 ± 6.93	16.33 ± 3.22	7.67 ± 4.04
Bho-P2-B2-S1-51	13.00 ± 5.57	5.00 ± 4.36	19.33 ± 4.73	1.33 ± 0.58
Nat-P3-M1-S1-79	12.67 ± 2.08	6.67 ± 2.89	14.00 ± 4.36	11.33 ± 4.73
Lak-P1-M1-S1-85	10.33 ± 1.53	5.00 ± 3.61	12.33 ± 1.53	1.67 ± 0.58
Lak-P1-S1-M1-46	9.67 ± 3.51	12.00 ± 5.20	11.67 ± 4.73	3.00 ± 3.00
Tha-P2-B1-S1-68	8.67 ± 1.16	5.33 ± 4.51	16.33 ± 7.02	11.67 ± 13.28
Nil-P2-B1-M1-36	8.33 ± 1.53	7.33 ± 0.58	13.67 ± 3.79	5.00 ± 2.65
Pan-P1-B1-S1-69	7.67 ± 2.08	6.33 ± 4.73	12.33 ± 3.22	1.67 ± 2.89
Din-P2-M1-M1-25	7.33 ± 2.08	13.00 ± 2.65	11.00 ± 3.00	14.33 ± 8.08
Mym-P2-M3-S1-45	7.00 ± 4.00	14.33 ± 16.65	14.33 ± 3.79	8.00 ± 4.36
Mym-P3-M2-S1-40	6.67 ± 1.16	4.00 ± 1.73	13.00 ± 6.93	10.67 ± 11.72
Noa-P1-B1-M1-31	4.00 ± 1.00	1.00 ± 1.73	13.00 ± 1.00	7.00 ± 4.00
Bog-P3-B1-S1-29	1.00 ± 0.00	6.67 ± 7.64	10.33 ± 3.06	13.00 ± 11.27
Tan-P1-B2-S1N-84	0.00 ± 0.00	0.00 ± 0.00	0.33 ± 0.58	11.00 ± 12.29

**Table 3 microorganisms-10-02282-t003:** Biomass dry weight (DW) and ARA activity in soybean Enrei variety and Binasoybean-3 variety inoculated with isolated bacteria and *B. diazoefficiens* USDA110. All experiments were performed in triplicate and the data are expressed as the mean ± STDEV.

	Enrei Variety		Binasoybean-3 Variety	
Bacteria Name	Biomass DW (g)	ARA (µmol/h/g Nodule DW)	Biomass DW (g)	ARA (µmol/h/g Nodule DW)
USDA110	0.95 ± 0.04	22.37 ± 20.18	0.52 ± 0.04	35.83 ± 12.78
Bho-P2-B2-S1-51	0.87 ± 0.20	109.76 * ± 7.04	0.57 ± 0.04	86.77 * ± 27.27
Bog-P3-B1-S1-29	0.96 ± 0.05	50.83 ± 88.41	0.50 ± 0.02	44.98 ± 11.93
Din-P2-M1-M1-25	1.02 ± 0.02	21.74 ± 29.17	0.46 ± 0.12	42.59 ± 31.99
Lak-P1-M1-S1-85	0.96 ± 0.06	18.71 ± 28.87	0.54 ± 0.07	64.66 ± 16.53
Lak-P1-S1-M1-46	0.95 ± 0.06	44.53 ± 41.29	0.49 ± 0.07	24.95 ± 12.07
Mym-P2-M3-S1-45	0.93 ± 0.03	30.92 ± 27.52	0.52 ± 0.05	21.80 ± 9.19
Mym-P3-M2-S1-40	0.89 ± 0.04	60.12 ± 20.58	0.50 ± 0.04	28.98 ± 7.37
Nat-P3-M1-S1-79	0.95 ± 0.11	12.53 ± 21.65	0.51 ± 0.10	69.23 ± 21.22
Nil-P2-B1-M1-36	0.98 ± 0.02	36.72 ± 39.18	0.60 ± 0.00	19.55 ± 11.68
Noa-P1-B1-M1-31	0.92 ± 0.07	5.01 ± 4.48	0.57 ± 0.06	24.58 ± 2.93
Pan-P1-B1-S1-69	0.79 ± 0.01	60.53 ± 60.33	0.59 ± 0.04	48.34 ± 21.62
Tan-P1-B2-S1N-84	0.85 ± 0.08	0.00 ± 0.00	0.50 ± 0.04	6.43 ± 5.84
Tha-P2-B1-S1-68	0.96 ± 0.08	30.77 ± 51.64	0.57 ± 0.05	13.20 ± 6.80

* Denotes significance with “*B. diazoefficiens* USDA110” at 95% confidence level using Dunnett’s test.

**Table 4 microorganisms-10-02282-t004:** Data of stress-tolerance tests of selected isolates.

	NaCl					Temperature						pH				
	0%	1%	2%	3%	4%	10 °C	20 °C	28 °C	37 °C	40 °C	45 °C	4.5	6	7	8.5	10
USDA110	(+)	(+)	(+)	(+)	(−)	(−)	(+)	(+)	(+)	(−)	(−)	(+)	(+)	(+)	(+)	(+)
Bho-P2-B2-S1-51	(+)	(+)	(+)	(+)	(+)	(−)	(+)	(+)	(+)	(+)	(+)	(+)	(+)	(+)	(+)	(+)
Bog-P3-B1-S1-29	(+)	(+)	(+)	(+)	(+)	(−)	(+)	(+)	(+)	(+)	(+)	(+)	(+)	(+)	(+)	(+)
Din-P2-M1-M1-25	(+)	(+)	(+)	(+)	(+)	(−)	(+)	(+)	(+)	(+)	(−)	(+)	(+)	(+)	(+)	(+)
Lak-P1-M1-S1-85	(+)	(+)	(+)	(+)	(+)	(−)	(+)	(+)	(+)	(+)	(−)	(+)	(+)	(+)	(+)	(+)
Lak-P1-S1-M1-46	(+)	(+)	(+)	(+)	(+)	(+)	(+)	(+)	(+)	(+)	(−)	(+)	(+)	(+)	(+)	(+)
Mym-P2-M3-S1-45	(+)	(+)	(+)	(+)	(+)	(−)	(+)	(+)	(+)	(+)	(+)	(+)	(+)	(+)	(+)	(+)
Mym-P3-M2-S1-40	(+)	(+)	(+)	(+)	(+)	(−)	(+)	(+)	(+)	(+)	(−)	(−)	(+)	(+)	(+)	(+)
Nat-P3-M1-S1-79	(+)	(+)	(+)	(+)	(+)	(−)	(+)	(+)	(+)	(+)	(+)	(+)	(+)	(+)	(+)	(+)
Nil-P2-B1-M1-36	(+)	(+)	(+)	(+)	(+)	(−)	(+)	(+)	(+)	(+)	(−)	(+)	(+)	(+)	(+)	(+)
Noa-P1-B1-M1-31	(+)	(+)	(+)	(+)	(+)	(−)	(+)	(+)	(+)	(+)	(−)	(−)	(+)	(+)	(+)	(+)
Pan-P1-B1-S1-69	(+)	(+)	(+)	(+)	(+)	(+)	(+)	(+)	(+)	(+)	(−)	(+)	(+)	(+)	(+)	(+)
Tan-P1-B2-S1N-84	(+)	(+)	(+)	(+)	(+)	(+)	(+)	(+)	(+)	(+)	(−)	(+)	(+)	(+)	(+)	(+)
Tha-P2-B1-S1-68	(+)	(+)	(+)	(+)	(+)	(−)	(+)	(+)	(+)	(+)	(+)	(+)	(+)	(+)	(+)	(+)

(+) denotes growth, (−) denotes no growth.

**Table 5 microorganisms-10-02282-t005:** Highest similarity of genes sequences of selected isolates with type strains.

Bacteria Name	16S rRNA Gene Similarity	ITS Region Similarity	23S rRNA Gene Similarity	Symbiotic Genes Similarity		House-Keeping Gene Similarity			
				*nifH* Gene Similarity	*nodD1* Gene Similarity	*atpD*	*glnII*	*gyrB*	*MLSA*
Bho-P2-B2-S1-51	*B. liaoningense*	*B. liaoningense*	*B. liaoningense*	*B. diazoefficiens*	*B. diazoefficiens*	*B. diazoefficiens*	*B. guangxiense*	*B. liaoningense*	*B. guangzhouense*
Bog-P3-B1-S1-29	*B. elkanii*	*B. elkanii*	*B. elkanii*	*B. elkanii*	*B. zhanjiangense*	*B. japonicum*	*B. paxllaeri*	*B. elkanii*	*B. elkanii*
Din-P2-M1-M1-25	*B. diazoefficiens*	*B. diazoefficiens*	*B. diazoefficiens*	*B. diazoefficiens*	*B. diazoefficiens*	*B. diazoefficiens*	*B. diazoefficiens*	*B. diazoefficiens*	*B. diazoefficiens*
Lak-P1-M1-S1-85	*B. liaoningense*	*B. liaoningense*	*B. liaoningense*	*B. diazoefficiens*	*B. ottawaense*	*B. diazoefficiens*	*B. guangxiense*	*B. liaoningense*	*B. diazoefficiens*
Lak-P1-S1-M1-46	*B. elkanii*	*B. elkanii*	*B. elkanii*	*B. elkanii*	*B. zhanjiangense*	*B. japonicum*	*B. paxllaeri*	*B. elkanii*	*B. elkanii*
Mym-P2-M3-S1-45	*B. diazoefficiens*	*B. diazoefficiens*	*B. diazoefficiens*	*B. diazoefficiens*	*B. diazoefficiens*	*B. diazoefficiens*	*B. diazoefficiens*	*B. diazoefficiens*	*B. diazoefficiens*
Mym-P3-M2-S1-40	*B. diazoefficiens*	*B. diazoefficiens*	*B. diazoefficiens*	*B. diazoefficiens*	*B. diazoefficiens*	*B. diazoefficiens*	*B. diazoefficiens*	*B. diazoefficiens*	*B. diazoefficiens*
Nat-P3-M1-S1-79	*B. diazoefficiens*	*B. diazoefficiens*	*B. diazoefficiens*	*B. diazoefficiens*	*B. diazoefficiens*	*B. diazoefficiens*	*B. diazoefficiens*	*B. diazoefficiens*	*B. diazoefficiens*
Nil-P2-B1-M1-36	*B. diazoefficiens*	*B. diazoefficiens*	*B. diazoefficiens*	*B. diazoefficiens*	*B. diazoefficiens*	*B. diazoefficiens*	*B. diazoefficiens*	*B. diazoefficiens*	*B. diazoefficiens*
Noa-P1-B1-M1-31	*B. diazoefficiens*	*B. diazoefficiens*	*B. diazoefficiens*	*B. diazoefficiens*	*B. diazoefficiens*	*B. diazoefficiens*	*B. diazoefficiens*	*B. diazoefficiens*	*B. diazoefficiens*
Pan-P1-B1-S1-69	*B. liaoningense*	*B. liaoningense*	*B. liaoningense*	*B. diazoefficiens*	*B. ottawaense*	*B. guangzhouense*	*B. guangxiense*	*B. liaoningense*	*B. guangzhouense*
Tan-P1-B2-S1N-84	*B. yuanmingense*	*B. arachidis*	*B. japonicum*	*B. yuanmingense*	*B. yuanmingense*	*B. diazoefficiens*	*B. arachidis*	*B. diversitatis*	*B. diazoefficiens*
Tha-P2-B1-S1-68	*B. diazoefficiens*	*B. diazoefficiens*	*B. diazoefficiens*	*B. diazoefficiens*	*B. diazoefficiens*	*B. diazoefficiens*	*B. diazoefficiens*	*B. diazoefficiens*	*B. diazoefficiens*

**Table 6 microorganisms-10-02282-t006:** Results of regression analysis of different parameters of pot experiment.

Name	Dependent Variable	Independent Variable	Pearson Correlation/ Standardized Coefficient Beta	Significance (*p*-Value)
Enrei	Biomass DW	Nodule dry weight	0.344	0.026 *
		Nodule number	0.219	0.163
		Medium size nodule	0.032	0.841
		Small size nodule	0.251	0.108
		ARA (in µmol/h/g nodule DW)	−0.164	0.301
		ARA (in µmol/h/plant)	−0.097	0.539
	ARA (in µmol/h/plant)	Nodule dry weight	0.325	0.035 *
		Nodule number	0.240	0.125
		Medium size nodule	0.293	0.060
		Small size nodule	0.106	0.506
Binasoybean-3	Biomass DW	Nodule dry weight	0.337	0.029 *
		Nodule number	−0.088	0.578
		Medium size nodule	0.343	0.026 *
		Small size nodule	−0.323	0.037 *
		ARA (in µmol/h/g nodule DW)	−0.101	0.524
		ARA (in µmol/h/plant)	0.037	0.817
	ARA (in µmol/h/plant)	Nodule dry weight	0.358	0.020 *
		Nodule number	0.213	0.176
		Medium size nodule	0.431	0.004 *
		Small size nodule	−0.093	0.558

* Denotes significance at 95% confidence level. The correlation analysis and pot experiment results indicated that the isolated bacteria have showed better activity in the Bangladeshi local variety, Binasoybean-3 than the Japanese variety, Enrei. The reason for this could be the higher symbiotic affinity, compatibility and effectiveness between isolates and plant variety of same origin—Bangladesh.

## Data Availability

All relevant data can be accessed in the manuscript and Appendix A.

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
