# Peer review of "Genetic and Physiological Characterization of Soybean-Nodule-Derived Isolates from Bangladeshi Soils Revealed Diverse Array of Bacteria with Potential Bradyrhizobia for Biofertilizers"

_microorganisms, 2022, doi:10.3390/microorganisms10112282_

Round 1

Reviewer 1 Report

Authors of the manuscript „ Genetic and physiological characterization of soybean-nodule-2 derived isolates from Bangladeshi soils revealed temperature 3 tolerant potential Bradyrhizobia for practical application as biofertilizers” by the authors  Md Firoz Mortuza, Salem Djedidi, Takehiro Ito, Shin-ichiro Agake, Hitoshi Sekimoto, Tadashi Yokoyama, Shin Okazaki  and Naoko Ohkama-Ohtu.

All sections require minimal corrections according to the attached file.

Reviewer 2 Report

The manuscript titled: “Genetic and physiological characterization of soybean-nodule-2 derived isolates from Bangladeshi soils revealed temperature 3 tolerant potential Bradyrhizobia for practical application as bio-4 fertilizers” got a total of 84 isolates from the nodules of soybean plants. According to the 16S rRNA gene sequencing and phylogenetic analysis, it showed that those isolates were mostly belonging to the genus Bradyrhizobium. The pot experiment found that several isolates showed better activity than USDA110 and some of them showed high NaCl tolerance and high temperature resistance, which make them as eco-friendly biofertilizer for soybeans in salty and tropical regions.

Major comments:

1.      The ARA is from the medium size nodule or the small size nodule? Why Bho-P2-B2-S1-51 show significant higher ARA, but the plant biomass did not changed. Please explain the possibility.

2.      Why selected these 13 isolates for further studies of symbiotic performances and stress tolerance assays? Moreover, for the pot experiment, three replicates for each isolate is not enough.

3.      How many isolates in the present study have been reported or released in the NCBI data base? How many of them are novel species of rhizobium? Those information should be included in the manuscript.

4.      Some of the isolates perform differently in different soybean genotypes. Please explained the reasons. And if the bacteria performances depend on soybean genotypes, how to make them the prospective candidate isolates as biofertilizers for soybean production?

5.      In general, author have isolated many rhizobium, characterized them geneticly and investigated the symbiotic performances and stress tolerance of some of them. But what is the novelty of this study?

Minor concerning:

1.      In figure S3, Asterisks indicates what?

2.      Lacking Statistical analysis in M&M

3. The title is not well fit to the content of the manuscript.

Reviewer 3 Report

The study has some interesting and novel aspects which would be of interest to the international scientific community. However, this manuscript needs minor improvements.

The title: Genetic and physiological characterization of soybean-nodule-derived isolates from Bangladeshi soils revealed temperature tolerant potential Bradyrhizobia for practical application as biofertilizers, should be shortened and rewritten, to be more concise and specific. For example,  “temperature tolerant potential Bradyrhizobia” does not reflect all the potentials of isolates for practical application as biofertilizers, nor all the potentials of isolates examined in the study.

In the Abstract:

Line 23: “expedition” – find better word.

Line 30: “families” – species

Line 33: “cv.” - delete

In the Materials and Methods:

Data on statistical analysis should be included in this section!!!

Line 109: had been – has been

108-111: include more information about soil sampling (e.g. number of samples, samling method) and soil properties (e.g. chemical and microbial soil properties, if they are determined)

Line 116-117: Five-fold dilutions 116 (1 g / 5 mL) of soil suspensions were used as inoculants – unclear, more detail are needed.

Line 142, 147: colony morphology or bacterial growth?

Line 149: explain how isolates were grown before genomic DNA extraction

Line 172-189: include information about experimental design and repetitions

In the Results:

Table 2 and 3, Figures S3 and S4 – Explanation for values following means in tables, as well as for bars on the charts are missing. Authors should include letters to denote significance, not only with “B. diazoefficiens USDA110”, but also with each other.

In the Discussion:

Authors should cite other research in this area and connect this research to other research in a more meaningful way, especially in the parts that refer to symbiotic performances and stress tolerance.
